# The Indentation-Induced Pop-in Phenomenon and Fracture Behaviors of GaP(100) Single-Crystal

**DOI:** 10.3390/mi10110752

**Published:** 2019-11-02

**Authors:** Yi-Jui Chiu, Sheng-Rui Jian, Jyh-Wei Lee, Jenh-Yih Juang

**Affiliations:** 1School of Mechanical and Automotive Engineering, Xiamen University of Technology, Xiamen 361024, China; chiuyijui@xmut.edu.cn; 2Department of Materials Science and Engineering, I-Shou University, Kaohsiung 840, Taiwan; 3Department of Materials Engineering, Ming Chi University of Technology, New Taipei City 243, Taiwan; jefflee@mail.mcut.edu.tw; 4Center for Plasma and Thin Film Technologies, Ming Chi University of Technology, New Taipei City 24301, Taiwan; 5Department of Mechanical Engineering, Chang Gung University, Taoyuan 33302, Taiwan; 6Department of Electrophysics, National Chiao Tung University, Hsinchu 300, Taiwan; jyjuang@g2.nctu.edu.tw

**Keywords:** GaP(100) single crystal, pop-in, nanoindentation, fracture

## Abstract

The deformation behaviors and fracture features of GaP(100) single-crystal are investigated by using nano- and micro-scale indentation techniques. The hardness and Young’s modulus were measured by nanoindentation using a Berkovich diamond indenter with continuous contact stiffness measurements (CSM) mode and the values obtained were 12.5 ± 1.2 GPa and 152.6 ± 12.8 GPa, respectively. In addition, the characteristic “pop-in” was observed in the loading portion of load-displacement curve, which was caused by the nucleation and/or propagation of dislocations. An energetic estimation methodology on the associated nanoindentation-induced dislocation numbers resulting from the pop-in events was discussed. Furthermore, the Vickers indentation induced fracture patterns of GaP(100) single-crystal were observed and analyzed using optical microscopy. The obtained fracture toughness *K_C_* of GaP(100) single-crystal was ~1.7 ± 0.1 MPa·m^1/2^, which is substantially higher than the *K*_IC_ values of 0.8 MPa·m^1/2^ and 1.0 MPa·m^1/2^ previously reported for of single-crystal and polycrystalline GaP, respectively.

## 1. Introduction

The III–V zincblende GaP is one of the most widely utilized substrates for fabricating various semiconducting and magnetic thin films [1,2,3,4]. From the viewpoint of device applications, it is thus of crucial importance to fully recognize the response of substrates due to mechanical stresses introduced during fabrication processes, which might cause the contact-induced damage and cracking. Accordingly, in order to implementing the important functional components of the devices on the GaP substrates, an accurate measurement of its mechanical characteristics is highly demanded.

Nanoindentation has been proved to be a powerful tool for studying the fundamental mechanical properties (for instance, the hardness and Young’s modulus) of various nanostructured materials [5,6,7], biomaterials [8,9], and thin films [10,11,12,13,14]. Combining the nanoindentation with microscopic techniques, the nanoindentation-induced shear band formation [15,16,17] in bulk metallic glasses, dislocation nucleation [18,19,20] in GaN films, and phase transition [21,22,23] in single-crystals Si and Ge had been extensively investigated to reveal the underlying mechanisms in great detail. For instance, it was indicated that in many nanoindentation measurements, the materials often exhibits a characteristic feature called the “pop-in” event characterized by a sudden displacement burst at a nearly constant indentation load, which acts as a trigger signifying the onset of plastic deformation. On the other hand, the “pop-in” behaviors have been attributed otherwise to the crack nucleation and the delamination phenomena [24]. In some cases, it has been also interpreted as the manifestations of the dislocation activity [18,19,20] depending strongly on the crystal structure of the test materials [25], temperature [26], the shape of indenter tip [27] and indenter angle [28]. Furthermore, the correlations between pop-in behaviors and dislocation activities of materials have been widely studied by combining microstructural observations with the cross-sectional transmission electron microscopy in recent years [29,30,31].

For the nanoindentation responses, the single “pop-in” behavior is often observed in the zincblende-structured, such as GaAs and InP single-crystals [31,32]. Whereas the multiple “pop-ins” has been ubiquitously identified in the hexagonal-structured materials, such as GaN thin films [33] and ZnO single-crystal [29]. However, although pop-in phenomenon in the loading segment was cited in the zincblende-structured GaP single-crystal [34], the features were not as apparent, presumably due to the sensitivity of testing module used in their indentation measurements was force module, which often resulted in serrated loading curve and blurred the feature of pop-in events. Herein, in this study, the mechanical properties and indentation-induced deformation behaviors of GaP(100) single-crystal were investigated by using continuous stiffness measurements (CSM) nanoindentation module to unveil the fundamental aspects of plasticity and mechanisms of local dislocation activities. Using the classical dislocation theory [35], we also estimated the number of indentation-induced dislocation loops formed at the initial stage of nanoscale deformation in GaP(100) single-crystal. Moreover, the Vickers indentation-induced fracture behaviors and mechanisms of GaP(100) single-crystal will be discussed, as well.

## 2. Materials and Methods

The (100)-oriented single-crystal GaP (1 × 0.5 cm^2^ in size, 0.5 mm-thickness and surface roughness: 0.2 nm) was purchased from Semiconductor Wafer Inc (SWI, Hsinchu, Taiwan). The nanoindentation tests were conducted using the Nanoindenter MTS NanoXP^®^ system (MTS Cooperation, Nano Instruments Innovation Center, Oak Ridge, TN, USA) equipped with a continuous stiffness measurement (CSM) module [36], which was accomplished by superimposing a small oscillation on the force signal and measuring the displacement response at the same frequency of 75 Hz. A pyramid-shaped Berkovich diamond tip with a radius of curvature around 50 nm was used as the indenter. The indenter was set three times to confirm that the indenter tip was properly in contact with materials surface and that any parasitic phenomenon was released from the tests. Next, the indenter was loaded for the fourth and final time at a strain rate of 0.05 s^−1^ until reaching the indentation depth. At the peak load the tip was held for 30 sec to avoid the influence of creep on unloading properties, which were used to calculate the mechanical properties of GaP(100) single-crystal. Finally, the tip was withdrawn with the same strain rate until 10% of the peak load was reached. In each measurement, 20 indentations were performed and each indentation was separated by 20 μm to avoid any possible interference from the neighboring indents [37,38].

The analytic method of Oliver and Pharr [39] was adopted to calculate the hardness and Young’s modulus of GaP(100) single-crystal. The hardness is defined as the applied indentation load divided by the projected contact area; that is, $H=P_{m}/A_{C}$, where $A_{C}$ is the projected contact area between the indenter tip and the surface at a maximum indentation load, $P_{m}$. For a perfectly sharped Berkovich indenter tip, the projected area is given by $A_{c}=24.56\;h_{c}^{2}$ (where $h_{c}$ is the true contact depth). The elastic modulus of material can be further calculated based on the relationships developed by Sneddon [40], as following:(1)$$E_{r}=\frac{1}{2\beta }S{\left(\frac{\pi }{A_{c}}\right)}^{1/2}$$
where, *S* is the contact stiffness of material and *β* is a geometric constant with *β* = 1.00 for Berkovich indenter. The reduced elastic modulus, $E_{r}$, can be calculated by using the equation:(2)$$\frac{1}{E_{r}}=\left(\frac{1-v_{i}^{2}}{E_{i}}\right)+\left(\frac{1-v_{\mathrm{GaP}}^{2}}{E_{\mathrm{GaP}}}\right)$$
where *ν* is the Poisson’s ratio, *E* is Young’s modulus. The subscripts “*i*” and “GaP” are indicated as the diamond indenter and GaP(100) single-crystal, respectively. For diamond indenter tip, *E_i_* = 1141 GPa, $\nu_{i}$ = 0.07 [39] and, $\nu_{\mathrm{Gap}}$ = 0.3 [41] is assumed for GaP(100) single-crystal. Further combining Equations (1) and (2), one can obtain the expression of Young’s modulus (*E*_GaP_) for GaP(100) single-crystal, as following:(3)$$E_{\mathrm{Gap}}=\frac{SE_{i}\left(1-v_{\mathrm{Gap}}^{2}\right)\sqrt{\pi }}{2\beta E_{i}\sqrt{A_{c}}-S\left(1-v_{i}^{2}\right)\sqrt{\pi }}$$

Therefore, the indentation displacement dependence of the hardness and Young’s modulus of materials can be obtained.

The Vickers indentations were made on GaP(100) single-crystal using a hardness tester (Akashi MVK-H11) to characterize the cracking behaviors. It is found that if indentation load is too high (e.g., 5 N) severe cracking and spallation will occur. As a result, in this study, the indentation load of 1.96 N was chosen, because it resulted in distinguishable crack pattern without spallations. Nine indents were performed at room temperature. The indentation-induced cracking patterns are observed by an optical microscope (OM) to obtain the indentation size and radial crack length along all directions statistically, which were used subsequently for calculating the fracture toughness of GaP(100) single-crystal.

## 3. Results

### 3.1. Nanoindentation

Ten CSM load-displacement curves of GaP(100) single-crystal reflecting the elastic and plastic behavior in course of nanoindentation is displayed in Figure 1. This curve clearly exhibits a displacement burst on the loading curve, known as the characteristic “pop-in” behavior, occurring at a critical load of about 0.9 mN, as shown in the insert of Figure 1. The pop-in behavior can be regarded as the manifestation of sudden activities of dislocations [32,42], giving rise to the seemingly discontinuous plastic deformation in the course of nanoindentation. Another noticeable feature in Figure 1 is that no discontinuities observed in the unloading curve, the so-called “pop-out” event. Generally, the “pop-out” event is a pressure-induced phase transition and has been observed in Si single-crystals [21,23,43] and Ge single-crystals [22,44]. The absence of this event indicates that there might not be any phase transition involved in the present case.

The curves of hardness and Young’s modulus versus the indentation depth obtained from the CSM analyses of GaP(100) single-crystal are plotted in Figure 2. Both curves exhibit a similar tendency, namely an initial quasi-linear increase to a maximum value, followed by a subsequent sharp decrease and then finally approaches to a constant value. It is interesting to note that the sharp decrease after the stage essentially coincides with the pop-in event, indicating a bursting activity of dislocation. The fact that the curves of hardness and the Young’s modulus both reaching constant values at a moderate penetration depth ensures the reliability of measurements, and the values of hardness and Young’s modulus of GaP(100) single-crystal obtained at this stage can be regarded as its intrinsic properties. In this study, the hardness and Young’s modulus of GaP(100) single-crystal obtained are about 12.5 ± 1.2 GPa and 152.6 ± 12.8 GPa, respectively. The value of Young’s modulus obtained in this work is consistent with the value of 147 ± 5 GPa reported by Grillo et al. [34]. However, the hardness reported in [34] was highly dependent on the maximum indenting force applied, with the values of 12.5 ± 0.2 and 10.9 ± 0.2 for the applied load of 1 and 10 mN, respectively. Since the CSM measurement instantaneously reflects the depth-dependence of hardness and Young’s modulus, thus the fact that both the hardness and Young’s modulus reaches its intrinsic values (as displayed in Figure 2) at an indentation depth of about 20 nm with a corresponding applied load of about 0.2 mN (see the inset of Figure 1) indicates the value of the intrinsic hardness of GaP should be more likely 12.5 ± 1.2 GPa. It is noted that although previous studies [25,45,46,47] have indicated that the difference of hardness and Young’s modulus values obtained in various semiconducting materials could be attributed to the crystalline orientation of thin films or single crystals, it is also plausible to deduce that the different operation modes of the nanoindenter or the indented plane may lead to the dissimilar results. To illustrate this point, values of hardness and Young’s modulus of various semiconductor single-crystals and thin films are listed in Table 1. In the future, we do anticipate that more details of mechanical properties and microstructural deformation mechanisms of (100)-, (110)- and (111)-oriented GaP single-crystals can be rigorously investigated in a comparative manner.

### 3.2. Homogeneous Dislocation Nucleation

In a previous study reported by Onodera et al. [53], the GaP zincblende phase undergoes a structural transition to a metallic phase II (*β*-Sn-type structure) as the pressure is higher than 20 GPa, which is higher than the hardness of GaP(100) single-crystal obtained in this work. Also, no characteristics of pressure-induced metallic phase transition phenomena can be observed in AFM image, as shown in the right hand side of Figure 1. The similar pop-in behavior has been investigated in the zincblende-structured GaAs and InP [32], indicating that the deformation mechanisms were indeed correlated to the massive dislocations nucleation and propagation during nanoindentation. Consequently, it is quite plausible to assume that similar behavior/mechanism also dominates in GaP(100) single-crystal.

According to the above-mentioned discussion, the pop-in appeared in the loading part naturally reflects the onset of nanoplasticity of GaP(100) single-crystal, which was manifested by the sudden nucleation and propagation of dislocations. That is, the corresponding indentation load is associated with the critical shear stress (*τ*_max_) and the energy connected with the “pop-in depth” may directly account for the number of the nanoindentation-induced newly nucleated dislocation loops. Following Johnson’s analytical model [54], the *τ*_max_ can be related to an indentation load (*P_c_*), as given below:(4)$$\tau_{\max}=\left(0.31/\pi \right){\left(6P_{c}E_{r}^{2}/R^{2}\right)}^{1/3}$$
where *R* is the radius of indenter tip. The *τ*_max_ for GaP(100) single-crystal thus obtained is about 3.8 GPa. Hence, the shear stress that initiates the plastic deformation and energy required for creating a dislocation loop to promote the deformation can be calculated. The free energy of a circular dislocation loop of radius (*r*), is given as:(5)$$U=\gamma_{\mathrm{dis}}2\pi r-\tau b\pi r^{2}$$
where *γ*_dis_, *b* (~0.385 nm) [55] and *τ* are the line energy of dislocation loop, the magnitude of Burgers vector and the external shear stress, respectively. The first term on the right-hand side of Equation (5) describes the energy needed to generate a dislocation loop in an initially defect-free lattice. The second term of Equation (5) is interpreted as the strain energy released via the work done by the applied stress, which, in effect, expands the dislocation loop over a displacement of one Burgers vector. The existence of a dislocation will generate lattice strain spreading in the vicinity of the dislocation for *r* > *r*_core_, and the corresponding line energy *γ*_dis_ is given by [35]:(6)$$\gamma_{\mathrm{dis}}=\frac{Gb^{2}}{8\pi }\left(\frac{2-v_{\mathrm{GaP}}}{1-v_{\mathrm{GaP}}}\right)\left[ln\left(\frac{4r}{r_{\mathrm{core}}}\right)-2\right]$$
where *G* ≈ 49.7 GPa [55] is the shear modulus of GaP(100) single-crystal and, *r*_core_ is the radius of a dislocation core. By using Equation (6), the Equation (5) can be rewritten as:(7)$$U=\frac{Gb^{2}}{4}\left(\frac{2-v_{\mathrm{GaP}}}{1-v_{\mathrm{GaP}}}\right)\left(ln\frac{4r}{r_{\mathrm{core}}}-2\right)r-\tau_{c}b\pi r^{2}$$

Wherein the value of the critical resolved shear stress *τ_c_* is taken as half of *τ*_max_ [56]. Equation (7) is a function of the material properties and the free energy accounts for the dislocations generated during the pop-in event, which has a maximum value when the dislocation loops reaches at a critical radius, *r_c_*. The system gains energy when the dislocation loop radius, *r,* is larger than the critical radius. Equation (7) also indicates that this maximum energy decreases with increasing load and homogeneous creation of circular dislocation loop becomes possible without the thermal energy at *U* = 0 [57], which in turn triggers a pop-in event in the loading curve. With this condition and set *dU*/*dr* = 0 for a maximum yields: $\tau_{c}=2\gamma_{\mathrm{dis}}/br$ and $r_{c}=\left(e^{3}r_{\mathrm{core}}\right)/4$. By plugging in the associated numbers quoted above, one obtains *r*_core_ ≈ 0.19 nm and *r_c_* = 0.95 nm, respectively for GaP(100) crystal.

On the other hand, the number of dislocation loops generated in the pop-in can be calculated from the associated work-done, *W_p_*, during nanoindentation. *W_p_* is approximated as the product of critical loading (*P_c_*, which is about 0.9 ± 0.2 mN) and the sudden burst displacement (*d*_pop-in_, which is about 5.8 ± 0.4 nm), as shown in Figure 1. The zeroth order estimation gives *W_p_* ≈ 5.2 × 10^−12^ Nm, implying that ~10^6^ dislocation loops with the size of critical diameter are generated during the pop-in event. This number is relatively low and is in agreement with the scenario of homogeneous dislocation nucleation, instead of activated collective motion of pre-existing dislocations [58].

### 3.3. Fracture Toughness and Fracture Energy

The fracture toughness (*K_C_*) is another important mechanical property of a material, especially when considered to serve as the substrate for device fabrications. *K_C_* can be measured using the indentation-induced cracks at the corners of an indent through the Vickers indentation [59]. The typical pattern of the Vickers indentation-induced cracking on GaP(100) single crystal is displayed in Figure 3. The ratio of average cracking length (*l* = (*l*_1_ + *l*_2_)/4, where *l*_1_ and *l*_2_ are denoted in Figure 3) to the half-diagonal length of the indentation (*a*) obeys the criteria of Palmqvist cracks with 0.25 ≤ *l*/*a* ≤ 2.5. The following equation is thus used to calculate *K_C_* value of GaP(100) single crystal, which is reported by Niihara et al. [60,61], as:(8)$$K_{C}=\frac{0.009P}{a\sqrt{l}}{\left(\frac{E_{\mathrm{GaP}}}{H}\right)}^{2/5}$$
where *P* is the applied loading. The obtained *K_C_* value of the GaP(100) single crystal is about 1.7 ± 0.1 MPa·m^1/2^. Comparing with the reported *K*_IC_ values of single crystal GaP and polycrystalline GaP, which are 0.8 MPa·m^1/2^ and 1.0 MPa·m^1/2^ [62], respectively, the value obtained in the present study is substantially higher. Considering that *K_C_* is strongly related to materials geometry and is usually decreased with increasing sample thickness until reaching a minimum value known as *K*_IC_, we believe that the difference, although is noticeable, is in a reasonable range. The fracture energy (*G_C_*) of GaP(100) single crystal is also calculated using the equation: $G_{C}=K_{C}^{2}\left[\left(1-v_{\mathrm{GaP}}^{2}\right)/E_{\mathrm{GaP}}\right]$ [63]. The obtained value for fracture energy is approximately 23.8 J·m^−2^ in this study.

## 4. Conclusions

To sum up, the mainly findings of indentation-induced pop-in and fracture behaviors of GaP(100) single-crystal are summarized as following:From nanoindentation results, the obtained hardness and Young’s modulus of GaP(100) single crystal are 12.5 ± 1.2 GPa and 152.6 ± 12.8 GPa, respectively.The energetic estimation indicated that the number of dislocation loop is estimated to be in the order of 10^6^ for the pop-in event with a critical radius of 0.95 nm.The *K_C_* and *G_C_* values of GaP(100) single crystal obtained from the Vickers indentation test are about 1.7 ± 0.1 MPa·m^1/2^ and 23.8 J·m^−2^, respectively.

## Figures and Tables

**Figure 1 micromachines-10-00752-f001:**
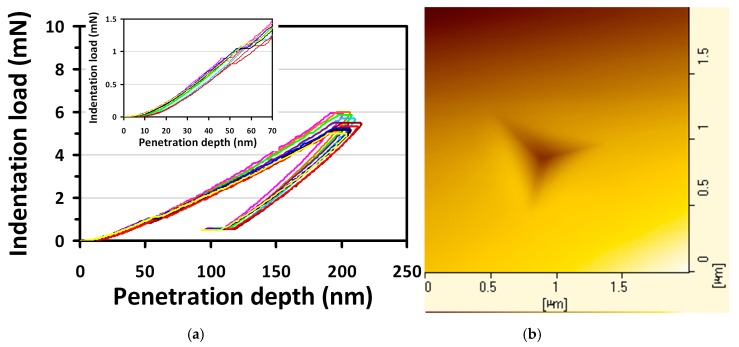
(**a**) Ten contact stiffness measurements (CSM) load-displacement curves show the “pop-in” events of GaP(100) single-crystal during nanoindentation. Insert: zoom in the range of penetration depth (0–70 nm) and indentation load (0–1.5 mN). (**b**) AFM image of the nanoindentation.

**Figure 2 micromachines-10-00752-f002:**
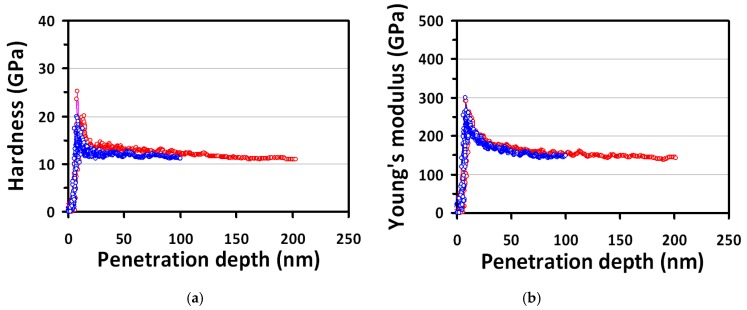
Hardness and Young’s modulus of GaP(100) single-crystal. (**a**) the hardness and (**b**) Young’s modulus as a function of penetration depth (100 and 200 nm) from nanoindentation CSM results.

**Figure 3 micromachines-10-00752-f003:**
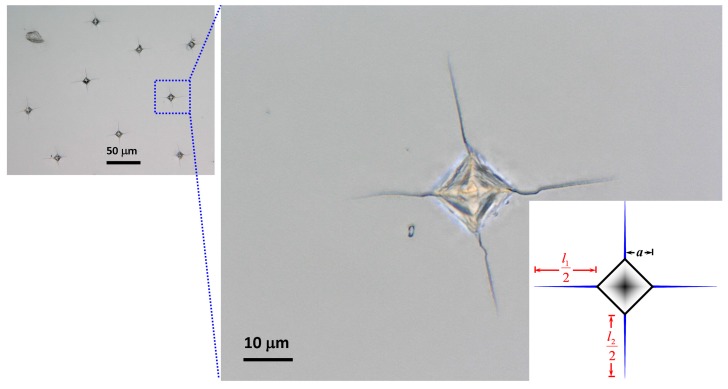
Palmqvist cracks obtained from Vickers indentation on GaP(100) single-crystal, where “*a*” is the half-diagonal of the indentation and “*l* = (*l*_1_ + *l*_2_)/4” is the average length of the radial cracks for each indentation under a load of 1.96 N.

**Table 1 micromachines-10-00752-t001:** The hardness and Young’s modulus of various semiconductor materials.

Materials	*H* (GPa)	*E* (GPa)
Single-crystal GaP(100) ^[#]^	12.5 ± 1.2	152.6 ± 12.8
Single-crystal GaP [34]	10.9 ± 0.2 (10 mN); 12.5 ± 0.2 (1 mN)	147 ± 5
Single-crystal GaAs(100) [32]	7.5	97
Single-crystal GaAs [34]	8.4 ± 0.1	123 ± 1
GaAs(100) thin films [48]	10.62 ± 0.3	118.97 ± 3.81
Single-crystal InP(100) [32]	5.1	82
InP layer [49]	6.83 ± 0.71	97.1 ± 0.68
InP substrate [49]	6.41 ± 0.25	90.12 ± 1.59
Single-crystal ZnSe [34]	1.47 ± 0.02	72 ± 2
ZnSe thin films [42]	2.0 ± 0.1	72.6 ± 0.5
Bulk InAs [50]	5	69.9
InAs layer [50]	7.8	100
Bulk ZnS [51]	1.9	75
ZnTe(111) thin films [52]	4	70

[#]: this study

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
