# Peer review of "The Indentation-Induced Pop-in Phenomenon and Fracture Behaviors of GaP(100) Single-Crystal"

_micromachines, 2019, doi:10.3390/mi10110752_

Round 1
Reviewer 1 Report
The manuscript was well written. However, there were some major technical issues to be resolved before it is publishable.
1) Line 78-79: In each measurement, 30 indentations were performed and each indentation was separated by 20um to
avoid any possible interference from the neighboring indents. Some relevant references should be added. For example,
T.F. Page, S.V. Hainsworth, D.T. Smith (Ed.), Proc. Int'l Conf. Instrumented Indentation, April 22–23, 1995, San Diego, CA, Special Publication of the National Institute of Standards & Technology, NIST, Gaitherburg, USA (1995)
2) Another pop-in at the peak load in Fig.1 was not discussed.
3) Fig.2 : The size dependent effect of Young's modulus seems to be resulted from the rip blunting relative to the shallow penetration. This has been discussed in [Journal of Materials Research 24 (2009), 590-598].
The authors also need to be aware that the tip get blunt quickly when indenting stiff and hard materials. Furthermore discussion should be made.
4) when using equation (8) to calculate the Kc, the authors should explain the values of E and H used. This is important because the measured apparent E and H clearly demonstrated the indentation depth dependent behaviour.
Furthermore, severe plastic deformation may also affect the applicability of equation (8). More detailed discussed were given in the following reference. (Journal of Physics D: Applied Physics 40 (2007), 5401)
Author Response
micromachines-615198
Dear Editor:
Enclosed please find the revised manuscript entitled “The indentation-induced pop-in phenomenon and fracture behaviors of GaP(100) single-crystal” prepared by Y.-J. Chiu, S.-R. Jian*, J.-W. Lee and J.-Y. Juang. In the revised version, we have made substantial modifications in accordance with most of the comments and suggestions raised by the reviewer. The instructive and rigorous comments from the reviewers are gratefully appreciated, as they have made significant improvement to our original manuscript.
A list of corrections and additions in responding to the reviewer’s queries is also attached for your editorial references. We appreciate your taking time in considering the publication of the current manuscript. Please inform me with any further instructions in this matter.
Best regards,
Sincerely yours,
Sheng-Rui Jian

Reviewer 2 Report
The authors present a nice set of micro- and nano-indentation experiments conducted on single crystal GaP(100), a material of significant commercial/technological interest in the semiconductor industry. The authors also present an extensive, detailed review and summary of the literature. The experiments appear to be well thought out and performed, although some important experimental parameters and considerations (e.g., sample size/thickness, surface roughness, choice of indentation loads, etc.) are not mentioned. The data presented is beautiful (e.g., perfect Berkovich indent in Fig 1, combined with a near textbook load-displacement curve containing a single visible pop-in event) and the authors appear to have concerned themselves with reproducibility by conducting replicate experiments. They also present reasonable arguments for why some of their measured values diverge from those in the literature. However, more discussion of the context, as well as the meaning and significance of their results, would greatly add to the impact of the work (i.e., what is the take-home message/why should the reader care?). Also, it is not clear exactly why the authors undertook this particular measurement given that values of the measured quantities for GaP were already present in the literature - e.g., is it particularly GaP(100) that is of interest (versus some other crystalline orientation), or is there some question as to the validity of literature values? The paper could benefit with a major revision in light of the above, with some minor copy editing of the English as well. More specific comments follow:
Fig 1 (line 117) shows a representative load displacement curve with a pop-in at 0.9 mN load indicated. Did all 30 nanoindentations (line 78) exhibit pop-ins? If so, at what load(s)? This is important in determining whether the subsequent values and conclusions derived re: dislocations are truly typical of single crystalline GaP(100). Likewise, it's not clear from the text whether the plots presented in Fig 2 of hardness and Young's modulus vs penetration depth are for the single load-displacement curve shown in Fig 1 (seems more likely), or are instead average values over the 30 indents performed. The reported hardness and Young's modulus values (line 131 and Fig 2, line 138) were determined by averaging over the values obtained in the depth range of 150-200 nm (line 129-130), yet the plots in Fig 2 clearly show that both the hardness and Young's modulus are still decreasing over this range, albeit much more gradually. Is there a reason the authors did not go to higher loads/larger depths (i.e., chose to use a max depth of 200 nm, per line 74, corresponding to ~6 mN load, per Fig 1/line 117, whereas in ref #34 the authors went up to loads of 10 mN)? How thick was the GaP crystal samples studied, and what was its surface roughness (to ensure the indents were significantly deeper than the surface roughness yet not so deep so as to result in significant substrate effects)? This needs to be reported and discussed. What is the significance of the calculated core and critical radii (line 185)? Are these sub-nm values reasonable? What do they suggest re: the dislocation cause and mechanism? How was the Vickers indentation load of 1.96 N chosen (line 99 and 213)? It seems a somewhat arbitrary value. What is the significance of the measured fracture toughness and energy? Please link this back to the motivation in the introduction (i.e., the use of GaP as a substrate for semiconductor devices and the types of mechanical stresses introduced during fabrication of said devices - lines 31-36). In other words, give the reader some context re: the meaning of the results. The same goes for the hardness and Young's modulus. How do these compare to other related semiconductor materials? Since the values were already in the literature (e.g., Ref #34), what motivated the authors to re-examine GaP? Based on the stated uncertainty (lines 131 and 218), the precision of the reported hardness and Young's modulus is perhaps best reported only to the nearest GPa.
Author Response

(The authors gave the same response as above.)

Round 2
Reviewer 1 Report
Now the manuscript has been improved a lot , which renders it to be accepted as it is.
Author Response
Enclosed please find the revised manuscript entitled “The indentation-induced pop-in phenomenon and fracture behaviors of GaP(100) single-crystal” prepared by Y.-J. Chiu, S.-R. Jian*, J.-W. Lee and J.-Y. Juang. In the second revised version, we thanks for the positive responses.
Reviewer 2 Report
The revised manuscript is markedly improved, and the authors have done a thorough job of responding to the comments and questions of both this and the other reviewer. In particular, the updated sample information (lines 73-74) along with the revised Fig 1 and Fig 2 data are particularly helpful in showing the reproducibility of the data and the soundness of the nanoindenation parameters chosen (e.g., max depth) to extract the quantities of interest (hardness and Young's modulus), which can be affected by surface roughness, sample thickness (substrate effects), and defects. It is now clear to the reader that the authors have taken all of this into consideration.
I believe a few minor revisions would still be of benefit prior to publication, as follows:
A brief (few sentence) description of the CSM technique (i.e., how a high frequency oscillatory load is superimposed on the quasi-static load to enable continuous depth-dependent stiffness measurements) would be beneficial for readers unfamiliar with the technique. This could occur around lines 66-67 where CSM is first introduced and/or lines 76-84 in the Materials & Methods section (where more details re: the CSM parameters employed in these experiments could be elaborated upon). In Fig 1, while the colors are very helpful in differentiating the 10 curves, employing slightly thinner lines for plotting the load-displacement curves (if possible) would make it easier to distinguish the individual curves. In lines 225-226, where the average values for the critical loading and sudden burst displacement values are given, it would be beneficial to also include standard deviations for those 2 measurements (to quantify the small spread in values evident from the 10 curves plotted in Fig 1). These averages (with standard deviations) could also perhaps be added to the list of conclusions (lines 261-266) to aid the reader. Likewise, a few more of the quantitative values summarized in the conclusions could perhaps be added to the abstract for ease of identification by readers ahead of reading the full paper. Finally, with regards to hardness/Young's modulus values of various semiconductors and crystal facets, The new Table 1 could perhaps be reorganized slightly so that values for a given material were ordered consecutively (e.g., single-crystal GaP and single-crystal GaP(100), followed by single-crystal GaAs then single-crystal GaAs (100) and GaAs(100) thin films) for ease of comparison. The authors could add a few sentences to the manuscript (building upon what was written in their reviewer response letter) re: future on other GaP single-crystal orientations and the motivation behind choosing to start with (100).Author Response
micromachines-615198
Dear Editor:
Enclosed please find the revised manuscript entitled “The indentation-induced pop-in phenomenon and fracture behaviors of GaP(100) single-crystal” prepared by Y.-J. Chiu, S.-R. Jian*, J.-W. Lee and J.-Y. Juang. In the second revised version, we have made substantial modifications in accordance with most of the comments and suggestions raised by the reviewer. The instructive and rigorous comments from the reviewers are gratefully appreciated, as they have made significant improvement to our original manuscript.
A list of corrections and additions in responding to the reviewer’s queries is also attached for your editorial references. We appreciate your taking time in considering the publication of the current manuscript. Please inform me with any further instructions in this matter.
Best regards,
Sincerely yours,
Sheng-Rui Jian
